# Controls of soil organic matter on soil thermal dynamics in the northern high latitudes

Dan Zhu [1], Philippe Ciais [1], Gerhard Krinner [2], Fabienne Maignan [1], Albert Jornet Puig[1] & Gustaf Hugelius[3,4]

Permafrost warming and potential soil carbon (SOC) release after thawing may amplify climate change, yet model estimates of present-day and future permafrost extent vary widely, partly due to uncertainties in simulated soil temperature. Here, we derive thermal diffusivity, a key parameter in the soil thermal regime, from depth-specific measurements of monthly soil temperature at about 200 sites in the high latitude regions. We find that, among the tested soil properties including SOC, soil texture, bulk density, and soil moisture, SOC is the dominant factor controlling the variability of diffusivity among sites. Analysis of the CMIP5 model outputs reveals that the parameterization of thermal diffusivity drives the differences in simulated present-day permafrost extent among these models. The strong SOC-thermics coupling is crucial for projecting future permafrost dynamics, since the response of soil temperature and permafrost area to a rising air temperature would be impacted by potential changes in SOC.

[1] Laboratoire des Sciences du Climat et de l'Environnement, LSCE/IPSL, CEA-CNRS-UVSQ, Gif Sur Yvette 91191, France. [2] CNRS, Univ. Grenoble Alpes, Institut de Géosciences de l'Environnement (IGE), Grenoble 38000, France. [3] Department of Physical Geography, Stockholm University, Stockholm 10691, Sweden. [4] Bolin Centre for Climate Research, Stockholm University, Stockholm 10691, Sweden. Correspondence and requests for materials should be addressed to D.Z. (email: dan.zhu@lsce.ipsl.fr)

Soils in the northern permafrost region contain ~1300 ± 300 Pg of soil organic carbon (SOC)[1–3], more than one-third of the global total SOC[4], formed under cold climates with limited decomposition. Carbon release from this large pool due to warming-induced thawing and acceleration of microbial decomposition may act as an important positive feedback to climate change, but the magnitude and timing of this carbon release remains uncertain[3,5–7]. Model estimates of changes in permafrost extent and carbon stocks vary widely for the recent decades[8] as well as for future scenarios[9,10], partly because different representations of soil thermodynamics lead to different soil temperature responses to air temperature changes. For the ground surface, the energy budget is composed of net radiation, sensible heat, latent heat from evapotranspiration, and heat flux into and out of the ground. On the other hand, below the surface, thermal dynamics is dominated by heat conduction[11] driven by surface soil temperature variations, and also affected by latent heat released or absorbed during soil water phase changes which can be important for wet soils in permafrost regions[12].

The heat conduction in the soil, assuming a uniform and constant medium, can be described by the one-dimensional Fourier's Law (Eqs. [1] and [2] in "Methods" section)[13,14], in which thermal diffusivity, $D$, is the key parameter that describes the rate at which soil temperature changes given a temperature gradient. The value of $D$ is determined by soil composition (minerals, air, water/ice, organic matter) and soil structure[11]. Among the different soil constituents, thermal diffusivity of soil organic matter (SOM) is an order of magnitude smaller than that of typical soil minerals, and slightly smaller than water[11]. Structurally, a higher organic matter content in the soil increases soil porosity[15], which also decreases soil thermal conductivity and $D$ especially when the soil pores are filled with air. As a result, SOM acts as an insulator and the presence of SOM cools the soil during summer, while its warming effect during winter is less important due to the insulating snow cover[16,17]. The summertime cooling effect of SOM in turn restrains microbial decomposition[5] and favours the accumulation of SOC. Given the high SOC storage in permafrost soils and the high vulnerability and sensitivity of these carbon stocks to projected warming[6,18], it is important to assess the quantitative impact of SOC on $D$ and the potential acceleration of soil warming due to the loss of SOC and its insulating effect.

Considering a harmonic forcing as the upper boundary condition (e.g., the seasonal cycle of surface soil temperature), the solution of the heat conduction equation (Eq. [4]) gives an exponential dampening of the temperature amplitude with increasing depth, with its e-folding depth reflecting the $D$ value[13,14] (Eq. [5]). Accordingly, $D$ can be derived from temperature amplitudes concurrently observed at two soil depths. This equation has been used to calculate $D$ and its influence factors in in situ studies[19–21], but these studies were limited to a few sites. A large-scale application of this equation has been applied by ref. [22] to study the snow insulation effect between air and soil temperature at 20 cm depth, but did not touch the research problem of heat conduction in sub-surface soils below 20 cm.

Here we infer $D$ from two large data sets of depth-specific soil temperature measurements, the Russian Historical Soil Temperature Data[23] (hereafter RHST) and the International Polar Year Thermal State of Permafrost[24] (hereafter IPY), at 274 locations that spread throughout Russia and North America including permafrost and non-permafrost regions (Supplementary Fig. 1). A key requirement for Eqs. (4) and (5) to be valid for inferring $D$ using real-world soil temperature profiles is that non-conductive processes, like the latent heat associated with soil freezing, have minor impacts on the heat budget of soils at

seasonal-cycle time-scale. Therefore, we examined the shape of temperature profiles at each site and selected only the sites where monthly temperature oscillations conform to sinusoidal functions, that is, to the solution (Eq. [4]) of heat conduction (see detailed data processing and discussion in Methods). After this data-filtering, 184 sites were retained. The calculated $D$ value using Eq. [5] can approximately represent the thermal diffusivity for the bulk soil in the corresponding depth interval at those selected sites. Then, we searched for emerging relationships between the observation-derived $D$ values and a suite of soil properties extracted from broad-scale soil maps, in order to identify the dominant factors that control $D$ and subsequently the propagation of temperature waves from ground surface to deeper soils. We find that SOC is the strongest predictor to explain the variability of $D$ among sites. Using the same method to calculate $D$ from CMIP5 model outputs, we find that the different parameterization of $D$ in these models dominates their large-scale performance in simulating today's permafrost extent. Further with a land surface model, we show that the thermal insulation of SOC increases modelled present-day permafrost carbon stock (0–3 m depth) by 39% ( + 230 Pg C) and permafrost area by 33% (+3.4 million km²), increasing agreement with broad-scale observational data. These findings demonstrate the need to explicitly represent the coupling between SOC and soil thermal dynamics in models that are utilized to project future permafrost changes.

## Results

**Organic matter content controls soil thermal diffusivity.** We find a significant negative correlation between soil thermal diffusivity (log-transformed) and soil organic carbon density (Fig. 1). This empirical result confirms that soils with more organic materials have a lower thermal diffusivity. Among all the tested soil characteristics including soil texture (fractions of sand, silt and clay)[25], bulk density[25], organic carbon content per unit dry weight[25] (g kg⁻¹), and soil volumetric moisture content[26,27] (Supplementary Figs. 2 and 3), the SOC density shown in Fig. 1

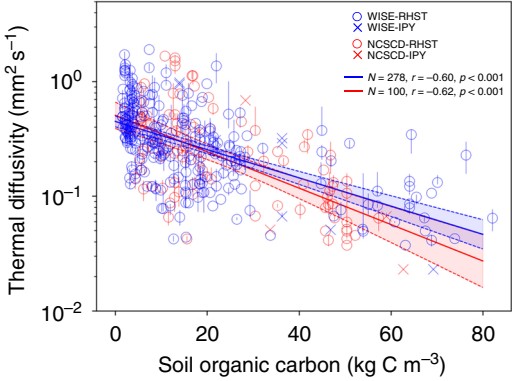

**Fig. 1** Relationship between soil thermal diffusivity and soil organic carbon (SOC). Thermal diffusivities ($D$) are derived from depth-specific measurements of monthly soil temperature from two data sets, RHST[23] (circles) and IPY[24] (crosses), at 184 sites (Supplementary Fig. 1), grouped into four depth intervals above 3 m (see Methods). The vertical error bars indicate 25th–75th percentiles among the available years for each site-depth. SOC densities (kg C m⁻³) are derived from two databases, WISE[25] (blue) and NCSCD[28] (red), integrated into the same four depth intervals. The blue and red solid lines indicate the linear regression lines of $\log_{10}(D)$ vs. SOC from WISE ($y = -0.012 \times -0.35$) and from NCSCD ($y = -0.016 \times -0.30$), respectively, with 95% confidence intervals shown in the dashed lines

has the strongest explanatory power for $D$ variations among the different site-depths. Considering the interrelation between these soil properties, e.g., SOC density is negatively correlated with bulk density (Supplementary Fig. 4), we conducted partial correlation analysis to test if the $D$-SOC relationship persists after removing the effect of other variables. The partial correlation between $D$ and SOC density remains significant after controlling the other properties separately or combined (Supplementary Table 1). This suggests that the impact of SOC on $D$ is a composite result of soil composition and structure alterations such that its correlation with $D$ cannot be fully accounted for by the variability in the texture- and porosity-related properties.

Uncertainties in the results shown in Fig. 1 and Supplementary Fig. 2 include a spatial-scale mismatch: the local soil temperature observations are not accompanied by collocated records of soil texture, moisture and organic carbon; we therefore resort to the global (WISE[25]) and regional (NCSCD[28]) databases for these soil properties, which are not fully representative for site-level observations. In addition, even at broad-scale, soil inventories bear substantial uncertainties[29,30]. The two databases used here differ in total carbon stock by about 20% for their overlapping areas (Supplementary Fig. 5ab, 690 Pg C in WISE vs. 825 Pg C in NCSCD for 0–2 m depth). Both databases derive their spatial distributions by linking field data of soil profiles to global/regional soil classification maps, their accuracy depending on both the accuracy of the up-scaling maps and the representativeness of samples within each soil class. Thus, the high landscape complexity in the northern permafrost zone and limited soil profiles available from Siberia[25,31] add to uncertainties in this region. To partly reduce this uncertainty, we excluded the site-depth data whose relative difference in SOC density between the two databases is higher than 25% (see Methods and Supplementary Fig. 5c).

Apart from the predominant role of SOC, soil moisture also affects $D$, especially in dry soils where an increased soil water act as bridges to improve thermal contacts between soil particles[11]. A laboratory experiment[32] showed that for sandy soils, $D$ increases rapidly with moisture content when the water content is less than 0.1–0.2 m³ m⁻³, and then remains stable or slightly decreases; while for cohesive soils of silty and clayey texture, the variation of $D$ with moisture is smaller. To investigate if soil moisture can explain the spatial variability of $D$, we used two data sets, the

satellite-based surface soil moisture product (ESA-CCI-SM[26]) and the depth-specific soil moisture simulated by a land surface model (ERA-Interim/Land[27]). Neither of them shows significant correlation with $D$, possibly because the annual mean moisture contents at the sites are mostly higher than 0.2 m³ m⁻³, a saturation level sufficient to bridge air gaps between soil grains (see Supplementary Fig. 3 and Supplementary Discussion). Apart from total moisture content, its phase composition also affects $D$, since ice has a much higher $D$ than water ($1.1$ vs. $0.14 \times 10^{-6}$ m² s⁻¹)[11]. Soils in the northern permafrost region often contain abundant ground ice[33] that can exceed soil porosity, including segregated ice and ice wedges[34]. These occurrences of excess ice beyond soil pores modify soil thermal properties and add to the heterogeneity of permafrost soils[35], contributing to the scatter in observations shown in Fig. 1 and Supplementary Figs. 2 and 3. Considering a potential confounding effect of ground ice on the correlation between SOC and $D$ shown in Fig. 1 (as a deeper depth could feature both lower SOC density and higher ice content, as well as a higher $D$), we calculated the regression for the top depth interval only (<0.4 m) (Supplementary Fig. 6), which is generally above the permanently frozen layer and less affected by large ice volumes. The similar results as Fig. 1 indicate that the observed strong correlation between SOC and $D$ is unlikely confounded by ice content.

**$D$ is a strong predictor for simulated permafrost extent.** Thermal diffusivity ($D$) is a key feature in the soil thermal regime. Its parameterization in land surface models may fundamentally impact their broad-scale patterns of soil temperature and permafrost distribution. Yet, this process has been given little attention. To test this, we analyzed the outputs of the historical experiment from a set of Earth System Models (ESMs) that participated in the fifth phase of the Climate Model Intercomparison Project (CMIP5)[36]. As the simulated climate differs among the models, the total area of grid cells with mean annual air temperature (MAAT) less than 0 °C ($A_{MAAT<0}$) ranges from 22 to 32 million km², with a standard deviation of 2.7 million km² (Fig. 2a). As expected, simulated near-surface permafrost area ($A_p$, defined as modelled active layer thickness less than 3 m (ref. [37]), see Methods) is significantly correlated with $A_{MAAT<0}$, with a slope slightly larger than 1, which might be related to the continuity of permafrost (namely, in models with a warm bias,

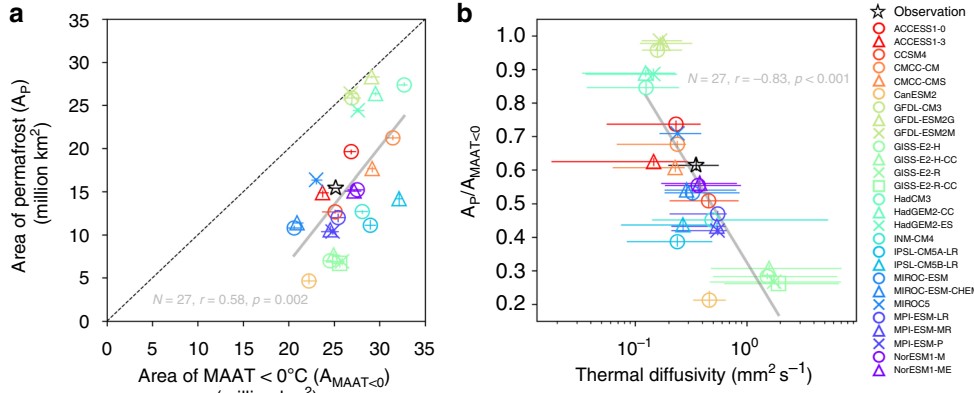

**Fig. 2** Relationship between permafrost area and thermal diffusivity in the CMIP5 models. **a** Modelled area of regions with surface mean annual air temperature (MAAT) less than 0 °C ($A_{MAAT<0}$) and area of near-surface permafrost ($A_p$, see "Methods" section) during the period 1961–1990, from the historical runs of CMIP5 models. Models from the same institute are shown in the same color. The error bars indicate 25th–75th percentiles among the 30 years. The black star indicates observations derived from CRU TS v3.24 climate dataset[65] and the IPA (International Permafrost Association) continuous and discontinuous permafrost extent[33]. **b** Area ratio between near-surface permafrost and regions with MAAT ≤ 0 °C ($A_p/A_{MAAT<0}$) during the period 1961–1990, against median thermal diffusivity from each CMIP5 model at the same site-years as the observations, with the horizontal error bars indicating 25th–75th percentiles. The solid grey line in (**b**) represents the linear regression line between $A_p/A_{MAAT<0}$ and $\log_{10}(D)$ for CMIP5 models ($y = -0.55 \times + 0.33$)

permafrost area is underestimated more strongly, as they miss the sub-grid discontinuous permafrost in the southern grid cells). However, the difference in MAAT among the models can only partly explain the much wider variation in $A_p$ across the models, with a standard variation of 7.5 million $km^2$ (Fig. 2a). The ratio between the two areas ($A_p/A_{MAAT<0}$) is determined by both air-soil interface energy exchanges affected by vegetation and snow covers, and sub-surface heat transfer in the soil column affected by soil thermal properties. Here it appears to be the latter that dominates $A_p/A_{MAAT<0}$ in the CMIP5 models, since 69% of the variation in $A_p/A_{MAAT<0}$ can be explained simply by differences in simulated $D$ between models (Fig. 2b). Note that the $D$ values for each model are diagnosed from their outputted monthly soil temperatures after excluding the pixels where non-conductive processes are not negligible (the same treatment as the observations, see Methods).

To further analyze whether simulated snow depth also controls $A_p/A_{MAAT<0}$, we calculated the effective snow depth[22] (the mean snow depth for each month weighted by its duration, see "Methods" section) for each model. In contrast to the strong correlation between $A_p/A_{MAAT<0}$ and $D$ (Fig. 2b and Supplementary Fig. 7a), effective snow depth is not significantly correlated with $A_p/A_{MAAT<0}$ among the CMIP5 models (Supplementary Fig. 7b). This is probably because of the following: (1) the insulation of snow occurs during winter, thus its effect on the maximum thaw depth is much smaller compared to the summer-time insulation by SOM, although the winter soil warming can be carried over to summer in regions with thick snow cover[38]; and (2) the CMIP5 models did not represent snow thermodynamics well (among these models, only CCSM4, MIROC and NorESM families used an explicit multilayer snow scheme[39]).

Interestingly, in Fig. 2b, the observation lies near the linear regression line of $A_p/A_{MAAT<0}$ against $D$ from the CMIP5 models, implying that the models generally have a reasonable structure for the soil thermal regime (most of the models adopt multilayer finite difference heat conduction[8]), but individual models have large biases of $A_p/A_{MAAT<0}$ related to their biases of $D$. This means that improving the parameterization of soil thermal properties should be a priority in order to capture permafrost distribution in these models. Note that most of the land models of CMIP5 ESMs did not consider the impact of SOM on soil physical properties[37]; nonetheless, modelled $D$ differs greatly among the models (Supplementary Fig. 8), possibly due to varied parameters for soil properties, different soil texture maps, and effects of different hydrological regimes.

**Thermal effect of SOC enhances soil carbon accumulation.** The reduction of soil thermal diffusivity by the presence of organic matter (Fig. 1) leads to a stronger dampening of the seasonal-cycle amplitude along the soil depth, and thus cools the deeper soils during summer, thins the active layer thickness (ALT), and reduces microbial respiration of SOC during the thawing season. To investigate the magnitude of this SOM effect on soil temperature and SOC accumulation, we conducted two simulations of the ORCHIDEE-MICT[40] land surface model, with (hereafter yesSOM) and without (hereafter noSOM) the impacts of SOM on soil thermo- and hydrodynamics. The SOM insulation effect in the model was implemented via SOM-dependent soil physical parameters including thermal conductivity, heat capacity, porosity and available water capacity, with detailed descriptions in ref. [40]. The inclusion of SOM effect allows the model to reproduce the relationship between SOC density and $D$, and to lower $D$ values which are more consistent with observations than the results of noSOM (Supplementary Fig. 9). Peak summer-time monthly mean soil temperature decreases by about 4 °C on

average at 0.8 and 1.6 m depths for the sites in permafrost region in yesSOM compared to noSOM (Supplementary Fig. 10). Consequently, ALT decreases by 0.6 m averagely across the region due to the presence of SOM (Fig. 3a, b), and the permafrost area (ALT < 3 m) increases from 10.4 in noSOM to 13.8 million $km^2$ in yesSOM, closer to the empirical estimate of 15.3 million $km^2$ for the continuous and discontinuous categories[33].

For SOC, modelled total carbon stock for 0–3 m depth in the northern permafrost zones (22 million $km^2$, according to IPA[33] including all permafrost categories) increases by 39%, from 590 Pg C in noSOM to 820 Pg C in yesSOM (Fig. 3e, f and Supplementary Fig. 11), which is still less than the estimate of 1030 Pg C in NCSCD database[1] over the same region. This discrepancy may in part be attributed to peat formation not being included in the current model[40]. The accumulation of SOC depends on both carbon input rate controlled by vegetation productivity, and carbon decomposition rate regulated by many abiotic (e.g., temperature, moisture, oxygen) and biotic factors[41]. Modelled gross and net primary productivity (GPP and NPP) is generally higher in yesSOM (Supplementary Fig. 12), since a higher water holding capacity due to the existence of SOM[42] can alleviate plant water stress during growing season. The relative increase of total primary productivities for the permafrost region is 8% (9.5 vs. 8.8 Pg C per year for GPP; 5.3 vs. 4.9 Pg C per year for NPP); thus, the 39% increase of SOC in yesSOM is mainly driven by the cooler summer-time soil temperature that slows down soil respiration in the model (Supplementary Discussion).

## Discussion

The key finding of this study is the dominant (spatial) impact of soil organic matter on soil thermal diffusivity ($D$) supported by empirical evidence, which is shown in model simulations to affect the broad-scale soil thermodynamics and in turn affects the soil organic carbon (SOC) accumulation. Thermal mediation of soil organic layer is known to stabilize the underlying permafrost in relatively warm regions where the climate alone cannot sustain the persistence of permafrost[43]. Here we provide a quantitative empirical estimate for the insulation effect of SOC through its impact on $D$. We recognize that, apart from heat conduction controlled by $D$, the latent heat associated with soil water/ice phase changes could be substantial in the soil energy budget, especially on the border of permafrost zones (Supplementary Fig. 1), which complicates the permafrost stability. This latent heat could partly explain the slower warming rate of the mean annual permafrost temperature in discontinuous zones during the past decades[24,44,45]. However, this latent heat effect delays but does not prevent the eventual permafrost thawing[45]. Under the long-term climate warming, as permafrost might greatly retreat poleward[39], heat conduction would finally dominate, inferred from the cold sites outside the permafrost regions today (Supplementary Fig. 1); and the SOC-$D$ relationship suggests a significant impact of changes in SOC on soil temperature in the former permafrost regions.

The implication of such SOC-$D$ feedback for future permafrost dynamics, therefore, depends on the direction and magnitude of soil carbon changes in the future. Although enzyme kinetics theory and short-term observations[5,46] suggest an initially increasing SOC decomposition rate with warming, large uncertainties still remain in the long-term response of soil carbon to climate change, due to uncertainties in the rate of SOC formation (including changes of plant productivity, soil-aggregate formation, etc.)[4,47] and in microbial physiological changes/adaptation to warming[48]. A recent compilation of field warming studies[49] showed diverging changes in SOC stock of top 10 cm depth after warming, with statistically neutral change across all studies. These

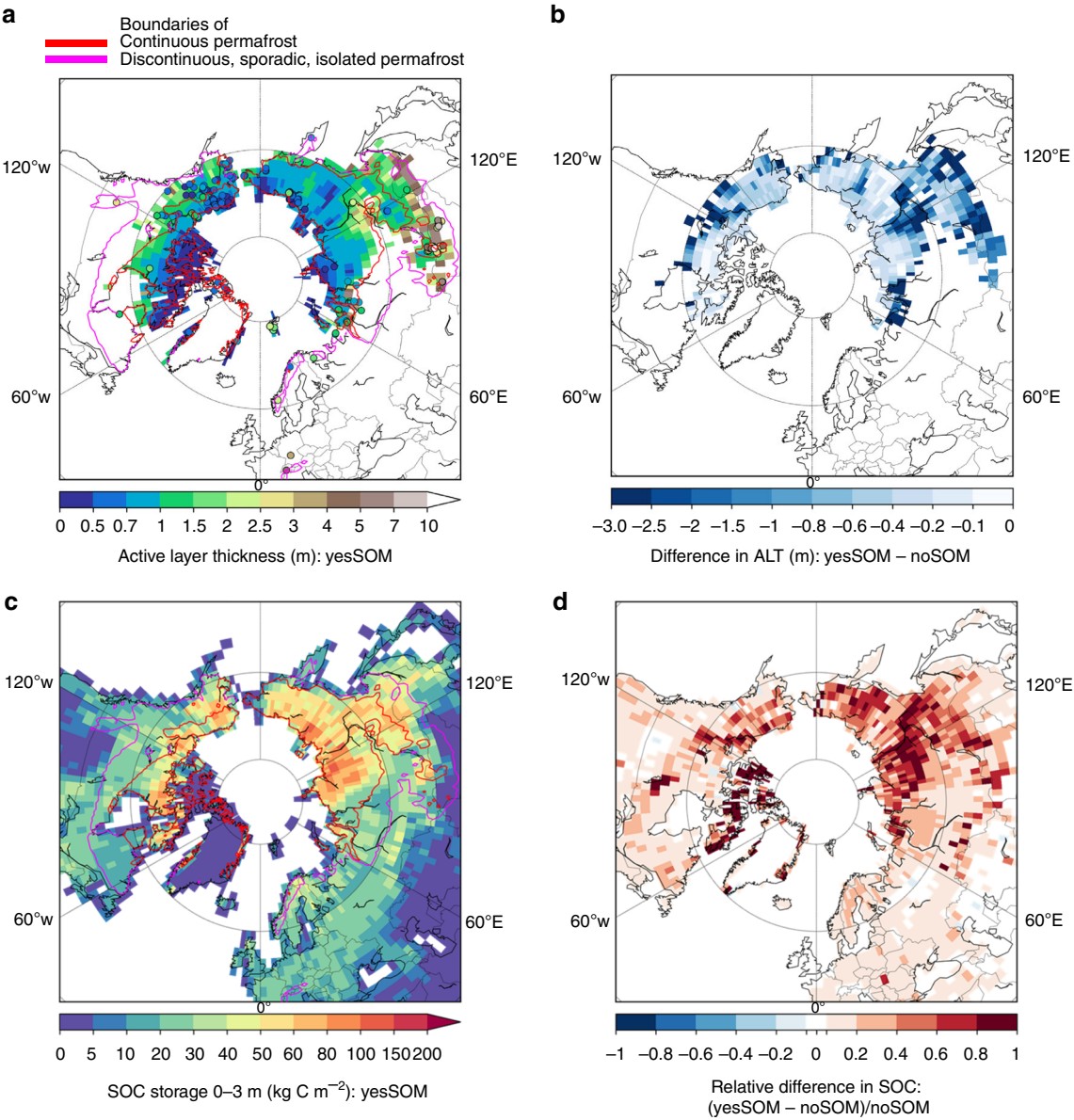

**Fig. 3** Impact of organic matter on modelled active layer thickness (ALT) and soil organic carbon (SOC). **a, b** Modelled mean ALT during 1990–2010 from yesSOM (**a**) and the difference between yesSOM and noSOM (**b**). The color-filled circles on (**a**) indicate observations from the CALM network. **c, d** Modelled mean SOC during 1990–2010 from yesSOM (**c**), and its relative difference against noSOM (**d**). The red and magenta lines indicate boundaries of northern permafrost according to the IPA map[33]

studies are, however, limited by a generally shallow depths of warming treatment, which missed the response of subsoils[50]; as well as a paucity of measurements for SOC stock changes at deeper soil horizons[6], especially for the formerly frozen soils. Laboratory incubation experiments, though, have demonstrated a high lability of permafrost soils under unfrozen conditions[51,52]. On the other hand, modeling studies mostly projected carbon losses in permafrost soils in the long run, albeit varying rates and magnitudes[3,53]. The protection of permafrost by SOC is also manifested in fire disturbances, where the combustion of surface organic layer led to increased soil temperature and deeper thaw depth[54,55]. Thus, the projected increase in fire activities in the northern high latitudes[56] might also be an important driver of permafrost thaw through the impact of SOC on soil thermal properties.

Apart from the dominant role of SOC in D, which controls the temperature propagation below ground (in the case of minor latent heat induced by freezing/thawing), the aboveground features including vegetation and snow cover also impact the vulnerability of permafrost to a rising air temperature by controlling heat exchanges between air and land surface[57]. Besides, degradation of ice wedges may lead to land subsidence and thus thermokarst formation, which is estimated to affect 20% of the northern permafrost region[58]. Such abrupt changes in the landscape strongly alter micro-topography and thus local hydrological and thermal regimes[59]. Meanwhile, the decadal trajectories after initial thaw are closely linked to accumulation of organic matter (itself related to vegetation succession) on the thermokarst troughs, which helps to stabilize ice wedges and prevent further thaw[60]. This mechanism, however, might be perturbed under future warming and intensified fire activities[61]. Such complex interactions call for models that quantitatively integrates thermo-hydrological processes, their heterogeneity at small spatial scales, vegetation dynamics and soil carbon cycle, thermal impacts of

organic matter, and fires, in order to improve our understanding and prediction of the overall vulnerability of permafrost.

The emergent slope of permafrost area versus $D$ from CMIP5 models (Fig. 2b) and the empirical relationship of $D$ versus SOC (Fig. 1) suggest a non-negligible sensitivity of permafrost area to potential SOC changes (a loss of ca. 0.2 million $\text{km}^{-2}$ per 1 kg C $\text{m}^{-3}$ reduction of SOC on average). Without the SOC-$D$ coupling, CMIP5 models have projected a linear relationship between air temperature change and permafrost area change[39]. The recent estimate[62] for the sensitivity of permafrost area to warming was also based on the assumption of an unchanged present-day MAAT-permafrost relationship for future scenarios. In contrast, the results shown here implies a non-linear trajectory of permafrost area changes in response to warming, highlighting a critical need to explicitly represent the coupling between SOC and soil thermal dynamics in models that are utilized to project the future of permafrost.

## Methods

**Calculation of $D$ from soil temperature measurements**. On the basis of Fourier's law, the one-dimensional heat conduction can be described by:

$$c\frac{\partial T}{\partial t} - \frac{\partial}{\partial z}\left(\lambda\frac{\partial T}{\partial z}\right) = 0, \tag{1}$$

where $T$ is temperature at time $t$ and depth $z$; $\lambda$ is thermal conductivity (J $\text{m}^{-1}$ $\text{s}^{-1}$ $\text{K}^{-1}$); $c$ is volumetric heat capacity (J $\text{m}^{-3}$ $\text{K}^{-1}$).

If $\lambda$ and $c$ are constants, Eq. [1] can be re-written as:

$$\frac{\partial T}{\partial t} - D\frac{\partial^2 T}{\partial z^2} = 0, \tag{2}$$

where $D$ is soil thermal diffusivity ($\text{m}^2$ $\text{s}^{-1}$), equalling to the quotient of $\lambda$ and $c$.

To solve Eq. [2], we consider the case that the upper boundary condition is given by a sine-wave temperature oscillation, which represents the diurnal or seasonal cycle of surface soil temperature driven by air temperature fluctuations:

$$T(0, t) = \overline{T_0} + A_0\sin(\omega t + \varphi), \tag{3}$$

where $A_0$ is amplitude of the temperature oscillation at the surface, equaling to half of the difference between warmest and coldest values; $\omega$ is its angular frequency, equalling to $\frac{2\pi}{86400 \times 365}$ if we consider the seasonal cycle in a year; $\varphi$ is its phase; $\overline{T_0}$ is the mean surface temperature in a wave period.

In the solution of Eq. [2], after a sufficient long time, the term that describes the transient disturbance since $t = 0$ becomes negligible (i.e., the effect of the initial temperature function is lost)[13]. Thus, the solution of Eq. [2] for the periodic steady-state is as follows[13,14]:

$$T(z, t) = \overline{T_0} + A_0 e^{-\sqrt{\frac{\omega}{2D}}z}\sin\left(\omega t + \varphi - \sqrt{\frac{\omega}{2D}}z\right). \tag{4}$$

Equation (4) infers an exponential dampening of the temperature amplitude with increasing depth[19,63]:

$$\frac{A_{z1}}{A_{z2}} = e^{(z2-z1)\sqrt{\frac{\omega}{2D}}} \tag{5}$$

where $A$ is amplitude of the temperature wave at depth $z1$ or $z2$.

Therefore, $D$ can be calculated from observed temperature amplitudes at two soil depths during the same period.

Equation (4) is an idealized mathematical model to describe the real-world soil thermal regime, its applicability requiring several necessary conditions: first, the heat transfer in the soil is dominated by conduction, so that convection, radiation, and particularly the latent heat release/absorption associated with phase transitions are of minor importance at the seasonal-cycle time-scale; second, the soil thermal properties are relatively constant over time and uniform along the considered depth interval; and third, soil temperature at the upper boundary follows a sinusoidal function. If the real soil thermodynamics at a site significantly violates any of the above conditions, it is unlikely that the soil temperature function $T(z,t)$ would take the simple sinusoidal form as Eq. [4]. We therefore assume that, if the measured soil monthly temperatures at two adjacent depths both conform to sine-waves (with a small error), the soil thermodynamics in this interval can be approximated by Eq. [2], and the calculated $D$ using Eq. [5] is a valid value that describes the thermal diffusivity for the bulk soil in this depth interval at the site.

In this study, we used two data sets of in-situ soil temperature measurements to calculate $D$: the Russian Historical Soil Temperature Data[23] (RHST), and the International Polar Year Thermal State of Permafrost[24] (IPY). RHST provides monthly soil temperature records at 263 sites since the 1880s until 1990, but the majority of data exist after 1960. Here we used the last ten years' data, from 1981 to 1990. RHST contains measurements at multiple depths for near surface (0.02–0.2 m) and deeper (mostly at 0.2, 0.4, 0.8, 1.2, 1.6, 2.4, and 3.2 m) soils, but

the near surface measurements were taken in bare areas whereas the deeper soil measurements were taken under natural cover of grasses and snow[64]; so we only used the data below 0.2 m in our analysis. The IPY soil temperatures were measured at various depths dependent on the site, mostly for the years 2007–2013. The daily observations in IPY dataset were aggregated to monthly means.

For both data sets, we conducted the following data processing:

First, only the sites with available monthly temperature records for at least two nearby depths (depth difference less than 1 m) and for at least 12 consecutive months were used (in total 274 sites, 219 from RHST and 55 from IPY, Supplementary Fig. 1).

Second, to determine whether the monthly soil temperatures evolve as a sinusoidal cycle, we first fitted the 12 months to a sine curve as Eq. [3] using non-linear least squares regression, and then calculated the sum of square error (SSE) after normalizing by amplitude such that the SSE for different depths are comparable:

$$\text{SSE} = \sum_{i=1}^{12}\left(\frac{T_i - \hat{T}_i}{\hat{A}}\right)^2, \tag{6}$$

where $T_i$ is the temperature for month $i$; $\hat{T}_i$ is the predicted temperature with the fitted sine curve; $\hat{A}$ is the amplitude of the fitted curve.

To derive an SSE threshold, we assume that the fluctuations of monthly air temperature, according to the CRU dataset[65], generally approximate sine-waves, and calculated their SSE values for the northern hemisphere (Supplementary Fig. 13). The 75th percentile of the SSE values for air temperatures, 0.34, was chosen to filter the soil temperature data: the observation for a site-depth-year is removed if its SSE is higher than 0.34. After this filtering, 170 sites from RHST and 15 sites from IPY remained. Note that the retained sites do not necessarily fulfil the SSE threshold for all years or depths; those depth-years that exceed the threshold were also removed before the calculation of $D$.

Most of the high SSE situations are associated with the so-called zero curtain effect, namely, the soil temperature stays near 0 °C during extended periods of freezing/thawing, maintained by the release/absorption of latent heat and by other non-conductive processes including vapor transport and evaporation/condensation[66]. Supplementary Fig. 14 presents four examples from the IPY dataset to illustrate the disparity or similarity between soil temperature observations and the theoretical sine-waves. Whenever prominent zero curtain occurs, the SSE value will be much higher than 0.34 (Supplementary Fig. 14ab). Note that the presence of zero curtain effects, although of smaller impact on the season cycle after the data screening by SSE threshold, may lead to an underestimation in the derived $D$ values.

In terms of the spatial distribution of removed sites, the sites located in warmer permafrost regions (discontinuous, sporadic and isolated permafrost zones) have a higher removal percentage (52%, 41 sites out of 79) than those in the continuous permafrost zone (35%, 28 out of 80) and in non-permafrost regions (17%, 20 out of 115) (Supplementary Fig. 1). This could be plausibly explained by the near-0 °C of the mean annual permafrost temperature in the relatively warmer permafrost regions, where a partial freezing/thawing at upper permafrost and inter-annual climate variability could induce significant amounts of latent heat[24]. A comparison of MAAT of retained vs. removed sites also shows significant differences in both permafrost regions (including continuous and discontinuous categories) and non-permafrost region, with the removed sites having a closer-to-zero MAAT than retained sites (Supplementary Fig. 15). We also compared the ground ice abundance (extracted from the IPA map[33], which gives three levels of abundance) to test whether the filtering would preferentially remove the sites with high ground ice contents. As shown in Supplementary Fig. 16, no significant difference was found between removed and retained sites. No significant difference in the surface soil moisture (extracted from ESA-CCI-SM v02.2 product[26]) was found between the removed and retained sites (not shown).

To further test if applying the SSE threshold can effectively filter out the cases where the phase change-induced latent heat cannot be neglected at monthly time-scales, we conducted analyses based on simulations of the ORCHIDEE-MICT land surface model[40] (see the last part of Methods).

Third, for the retained site-depth-year data after step ii), we calculated the $D$ value for each depth according to Eq. [5], using the observed temperature amplitude of this layer and of the adjacent upper layer, only if the two depths are nearby (<1 m). Since we try to correlate $D$ with soil properties extracted from the global/regional soil databases which have fixed/limited layers (see below), the derived $D$ values for different depths were integrated into four depth intervals: 0–0.4, 0.4–0.8, 0.8–2, and 2–3 m, taking the median $D$ if multiple depths are available within each interval. Finally, if multiple years of $D$ are available for a site-depth, their median value is used for the regression analysis against soil properties, while their 25th–75th percentiles are shown in Fig. 1.

**Soil databases**. In order to identify the dominant influential factors on $D$, and considering that the soil temperature data sets do not report collocated measurements of soil properties at the Russian meteorological stations in the RHST dataset or the boreholes in the IPY dataset, we seek to extract soil information for these sites from the spatially explicit global/regional soil databases. To our knowledge, there are four such soil databases that are recent and/or widely used: HWSD[67];

WISE30sec[25] (hereafter WISE); SoilGrids250m[68] (hereafter SoilGrids); and NCSCD[28]. Among them, WISE is developed based on HWSD, but includes updates on the mapping unit composition, adopts a consistent taxotransfer procedure, and provides information down to 2 m[25]. We therefore used WISE instead of HWSD in this study. The SoilGrids product, unlike the other three databases that rely on soil classification maps to derive spatially explicit information, predicts soil properties using statistical methods from a list of covariates (climate, vegetation, topography, lithological units, etc), and thus can generate high-resolution maps (250 m) dependent on the resolution of input covariates[68]. However, in a preprocessing, we found that SoilGrids gives a much higher SOC stock (in kg C m$^{-2}$) than other databases in most grid cells in the northern hemisphere, and gives a global total SOC storage of 2757 Pg C for 0–1 m and 4570 Pg C for 0–2 m (calculated using their 250 m resolution map). This total SOC storage is almost twice the estimate by the majority of previous studies (see a short review in ref[30].). Therefore, we did not use SoilGrids in this study.

Compared to WISE, NCSCD provides SOC stock down to 3 m, and is based on more soil samples in permafrost region than WISE[25]. However, NCSCD is a regional database for the northern permafrost zones only, and does not provide other properties like soil texture and bulk density. We therefore used both databases for SOC and used WISE for other soil properties. At their highest resolution (0.012° for NCSCD and 30 arc second for WISE), we took the grid cells whose center is nearest to the coordinates of each observation site. The two databases provide different depth intervals (4 intervals down to 3 m in NCSCD, and 7 intervals down to 2 m in WISE), and we integrated their original layers into the same four depth intervals as $D$ values mentioned above (0–0.4, 0.4–0.8, 0.8–2, and 2–3 m), using depth-weighted averages. WISE does not provide information below 2 m, so its variables for the 4$^{th}$ depth interval are non-valid value in all the regression analyses.

Using the large-scale soil databases to indicate local-scale observations introduces uncertainties. To reduce this uncertainty, we compared the SOC density for the available site-depths from the two databases (Supplementary Fig. 5c), and excluded those with a relative difference (calculated as $\left|\frac{SOC_{NCSCD}-SOC_{WISE}}{SOC_{NCSCD}+SOC_{WISE}}\right|$) larger than 25%. This filtering process removes 105 site-depth SOC values, leaving 100 pairs of $D$-SOC$_{NCSCD}$ and 278 pairs of $D$-SOC$_{WISE}$ as shown in Fig. 1, spanning 184 sites (170 sites from RHST and 14 sites from IPY, Supplementary Fig. 1 red). For consistency, the excluded site-depths were also removed from the analysis of the other soil properties as shown in Supplementary Fig. 2.

A linear relationship was found between log-transformed $D$ and SOC density (Fig. 1); we therefore used $\log_{10}(D)$ in all the correlation analyses. A log-transformation of $D$ also improves its normality. To explore the potential multivariate relationships, we conducted a stepwise regression analysis using SOC density (from WISE and NCSCD respectively) and all the other soil properties as predictive variables. In the case of SOC from NCSCD, none of the other variables are significant in the stepwise regression to explain $D$ variations, namely, the addition of other properties does not increase the explanatory power compared to the $\log_{10}(D)$-SOC$_{NCSCD}$ regression model. In the case of SOC from WISE, only silt fraction is significant, but R$^2$ only marginally increases from 0.35 to 0.37, and the slope of SOC$_{WISE}$ in the stepwise regression equation ($-0.010$) is only slightly different from that in the single-variable regression ($-0.012$).

**Treatment of CMIP5 models**. We used all CMIP5 ESMs that provide access to their historical simulation (r1i1p1) on the website https://esgf-node.ipsl.upmc.fr/search/cmip5-ipsl/ (accessed in July 2017) and that provide surface air temperature and vertically-resolved soil temperatures (in total 27 models). To calculate active layer thickness (ALT) of each grid cell, we firstly calculated the maximum monthly soil temperature ($T_{s,max}$) for each soil level, then took the deepest depth where $T_{s,max}$ was at or above 0 °C. If the depth of 0 °C was between two soil levels, a linear interpolation was used to locate the depth. Permafrost is defined to be present in a grid cell if ALT is less than 3 m or the deepest model soil level, following ref[37]. which gives an approximation of near-surface permafrost.

For the simulated thermal diffusivity, the same procedure as the calculation of $D$ for the sites was carried out using the outputted monthly soil temperatures by each model for all land grid cells and soil layers. Specifically, the same SSE threshold of 0.34 was adopted to exclude the pixel-depth-years whose monthly soil temperature oscillations do not conform to sine-waves. Since the CMIP5 models have different (or a lack of) parameterizations for the soil freezing process[37], their SSE values differ (Supplementary Fig. 17). In general, the models that do not consider the latent heat associated with water phase change (CMCC, IPSL, and MPI families) produce sine wave-shaped monthly soil temperatures and thus have smaller SSE values (Supplementary Fig. 17). Applying the SSE threshold before calculating $D$ values for each model could reduce the confounding effect of their different parameterizations for the freezing-induces latent heat.

The CMIP5 models have different vertical soil layers (Supplementary Fig. 18), and here we only calculated the $D$ values for the upper 3 m, and integrated them into the same four depth intervals as the observations. Then, to compare with the observation-derived $D$ values at the RHST and IPY sites, the same locations (nearest grid cell center to the site coordinates) were selected for each model, thus deriving the results shown in Fig. 2b and Supplementary Fig. 7.

For the simulated snow depth, in order to better describe its insulation effect, we calculated the effective snow depth ($S_{eff}$) defined as the mean snow depth weighted by its duration, following ref. [22]:

$$S_{eff} = \frac{\sum_{i=1}^{6} S_i w_i}{\sum_{i=1}^{6} w_i}, \tag{7}$$

where $i$ (1–6) represents the winter months from October to March; $w_i$ is a weighting factor that equals to 6 for October and decreases to 1 for March; $S_i$ is the mean snow depth for the month $i$. In this way, an earlier accumulation of snow, with its longer time period to insulate the soil during the whole snow season, has a larger weighting and yields a higher $S_{eff}$ than the case of a linearly accumulating snowpack. Note that this calculation did not account for the compaction/densification-induced increase in snow thermal conductivity that weakens its insulating effect during the late snow season, as demonstrated by the observed smaller temperature gradient (soil temperature at 20 cm depth minus air temperature) under a certain snow depth during the late snow season than that during the early season under the same snow depth[40].

Only 21 models out of the 27 models provided outputs for snow depth (or snow mass) at the time we accessed the CMIP5 repository. For the models that only provided snow mass but not depth (the MPI and HadGEM families), a snow density of 250 kg m$^{-3}$ was used for conversion[39].

**Simulations of the ORCHIDEE-MICT land surface model**. We incorporated the impacts of organic carbon on soil thermal and hydrological properties in the ORCHIDEE-MICT land surface model, assuming that soil physical properties are weighted averages of mineral soil and pure organic soil. Detailed descriptions can be found in ref. [40]. One modification compared to ref. [40]. is that the soil thermal conductivity is calculated as geometric average of mineral and organic soils (instead of arithmetic average in ref. [40]), which is more consistent with the empirical linear relationship between $\log_{10}(D)$ and SOC density (Fig. 1). We conducted two simulations for the northern hemisphere (>30°N) at 2° spatial resolution forced by CRUNCEP v8 climate: yesSOM, in which an observation-based SOC map was used in the thermal and hydrological modules to derive soil properties; and noSOM, in which the thermal and hydrological modules see zero SOC, i.e., no impacts of SOC on soil properties. For yesSOM, the observational SOC map was constructed from NCSCD[28] for permafrost regions and from WISE[25] for non-permafrost regions, using linear vertical interpolation to convert their original soil horizons into ORCHIDEE-MICT vertical layers[40]. Here we did not use the prognostically modelled SOC by the carbon module of ORCHIDEE-MICT, in order to exclude bias of the carbon cycle module for the analysis of SOM effect. For both simulations, spin-ups were first conducted forced by looped 1901–1920 climate and pre-industrial CO$_2$ level of 286 ppm, covering 200 years of the full model and 20,000 years of the soil carbon sub-model to reach equilibrium for the pre-industrial period. Afterwards, transient runs for 1861–2010 were conducted forced by historical climate and rising CO$_2$ concentrations.

We also made use of the simulations to test the validity of the SSE threshold, and to compare the temperature amplitude-derived diffusivity ($D$) against the real diffusivity as used in the one-dimensional Fourier's equation to solve for soil temperatures in the model. ORCHIDEE-MICT includes a soil-freezing parameterization, in which the latent heat of fusion is incorporated into an apparent heat capacity, so that Eq. (1) writes [69]:

$$c_{app}\frac{\partial T}{\partial t} - \frac{\partial}{\partial z}\left(\lambda\frac{\partial T}{\partial z}\right) = 0,$$

$$\text{in which } c_{app} = c - \rho L\frac{\Delta\theta}{\Delta T}, \tag{8}$$

where $\rho$ is ice density (kg m$^{-3}$); $L$ is latent heat of fusion (J kg$^{-1}$); $\Delta T$ equals to 2 K, corresponding to the freezing window between $-1$°C and 1°C in the model; $\Delta\theta$ equals to the total water content (m$^3$ m$^{-3}$) for the layer, providing the available amount to freeze/thaw in the freezing window.

Then, thermal diffusivities:

$$D_{with-fusion} = \frac{\lambda}{c_{app}}$$

$$D_{no-fusion} = \frac{\lambda}{c} \tag{9}$$

can be calculated for each time-step and averaged over the year to compare with the amplitude-derived diffusivity ($D$). Detailed parameterizations of $\lambda$ and $c$ in ORCHIDEE-MICT are given in ref. [40], which generally follows the empirical equations of ref. [70] to account for the impact of soil moisture on thermal conductivity $\lambda$.

As shown in Supplementary Fig. 19a, the relative difference of annual mean $D_{with-fusion}$ and $D_{no-fusion}$ ($1 - \frac{D_{with-fusion}}{D_{no-fusion}}$, which ranges from 0 to 1) are generally small (less than 30%) for those locations where modelled monthly soil temperatures conform to sine-waves (SSE < 0.34). This suggests that applying the SSE threshold can effectively exclude the cases in which the latent heat significantly changes the apparent thermal diffusivity. Note that some site-depths have high SSE values (and thus were excluded) even though they are less affected by soil freezing/thawing (Supplementary Fig. 19a), probably because the other conditions for the

validity of Eq. (4) are not met, e.g., the diffusivity varies too much over the year or along the considered depth interval, or the surface soil temperature itself does not conform to a sine-wave (due to, e.g., a non-sine shaped air temperature, or a strong winter insulation of snow causing asymmetric effects on the seasonal-cycle).

Supplementary Fig. 19b compares the amplitude-derived diffusivity ($D$) with $D_{with-fusion}$, which shows a good correlation between them ($R^2 = 0.85$), although $D$ is generally smaller than $D_{with-fusion}$ when their values are low. This suggests that the amplitude-derived diffusivities (after excluding high SSE values) are reasonable indicators for the internal parameterization of soil thermal properties in land surface models.

## Data availability

The data sets analysed in this study are available online: Russian Historical Soil Temperature Data: https://data.eol.ucar.edu/dataset/106.ARCSS078; borehole temperature measurements during the International Polar Year (IPY): http://gtnpdatabase.org/boreholes; WISE30sec v1.0 soil database: http://www.isric.org/explore/wise-databases; NCSCD soil database: https://bolin.su.se/data/ncscd/tiff.php; ESA CCI SM v02.2 product: http://esa-soilmoisture-cci.org/node/202; soil moisture from the ERA-Interim/Land product: https://www.ecmwf.int/en/forecasts//archive-datasets/reanalysis-datasets/era-interim-land; CMIP5 model outputs: https://esgf-node.ipsl.upmc.fr/search/cmip5-ipsl/. Other data that support the findings of this study, such as the ORCHIDEE-MICT model outputs, are available from the corresponding author on reasonable request.

## Code availability

The python script to calculate thermal diffusivity from soil temperature measurements, as well as the input data to run the script, are provided in a Supplementary zipped file. This file contains the source data underlying Figs. 1 and 2 and Supplementary Figs. 2 and 7.

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

## Acknowledgements

The authors acknowledge the financial support from the European Research Council Synergy grant ERC-SyG-2013–610028 IMBALANCE-P, and from the GAP project in the framework of French-Swedish research cooperation program.

## Author contributions

D.Z. performed the analysis and produced the first version of the manuscript. P.C and G.K. provided advice and discussion throughout the process. F.M. and A.J. helped with the development for the SOC-thermic coupling in the ORCHIDEE-MICT model. G.H. contributed to analyses of soil C data sets and the discussion. All authors contributed to the writing of the manuscript.

## Additional information

**Competing interests:** The authors declare no competing interests.

