## [Peer Review File · Nature Communications]

Reviewers' comments:

Reviewer #1 (Remarks to the Author):

The presence of soil organic carbon (SOC) reduces thermal diffusivity (D), a key parameter in the soil thermal regime, and consequently cools the deeper soils during summer. In this study, the authors derive D from depth-specific measurements of monthly soil temperature at about 200 sites in the high latitude regions. They find that, among the tested soil properties including SOC, soil texture, bulk density, and soil moisture, SOC is the dominant factor controlling the variability of D among sites. Analysis of the CMIP5 model outputs reveals that the parameterization of D dominates the large-scale performance of these models in simulating permafrost extent. I have some questions, but most are minor:

1. In my opinion, the title seems to be too vast and should be given some necessary qualifiers. There is a very important concept in geography, i.e., temporal and spacial scale. The carbon in permafrost deposits mostly began to accumulate in the late Pleistocene except that northern peatlands developed mostly in the last deglaciation, especially during the Holocene. And the manuscript only pay attention to the circumarctic, not including the Tibetan Plateau.
2. What's the linear regression equation in Fig. 2(a)? Why is the regression coefficient bigger than 1.0? What's the meaning/implication of this? Please explain the reasons.
3. The form of the fitted linear equations needs to be consistent in figures, with significant levels given.
4. Why is thermal diffusivity given in form of \log_{10} but not of real values in figures? The real values are more readable for readers.

Reviewers' comments:

Reviewer #1 (Remarks to the Author):

The presence of soil organic carbon (SOC) reduces thermal diffusivity (D), a key parameter in the soil thermal regime, and consequently cools the deeper soils during summer. In this study, the authors derive D from depth-specific measurements of monthly soil temperature at about 200 sites in the high latitude regions. They find that, among the tested soil properties including SOC, soil texture, bulk density, and soil moisture, SOC is the dominant factor controlling the variability of D among sites. Analysis of the CMIP5 model outputs reveals that the parameterization of D dominates the large-scale performance of these models in simulating permafrost extent. I have some questions, but most are minor:

1. In my opinion, the title seems to be too vast and should be given some necessary qualifiers. There is a very important concept in geography, i.e., temporal and spacial scale. The carbon in permafrost deposits mostly began to accumulate in the late Pleistocene except that northern peatlands developed mostly in the last deglaciation, especially during the Holocene. And the manuscript only pay attention to the circumarctic, not including the Tibetan Plateau.

Indeed, permafrost soil carbon has been accumulated at long time scales, during which the climate, vegetation, and hydrologic and sedimentary environment have varied. We agree that the previous title, 'controls...on permafrost carbon dynamics', is not appropriate. And it's also true that we did not include permafrost at Tibetan Plateau, meanwhile, some cold sites outside today's permafrost zones were included. We therefore revise the title as "*Controls of soil organic matter on soil thermal dynamics in the northern high latitudes*".

2. What's the linear regression equation in Fig. 2(a)? Why is the regression coefficient bigger than 1.0? What's the meaning/implication of this? Please explain the reasons.

In Fig. 2a, the x-axis reflects bias in the simulated present-day air temperature by CMIP5 models, while y-axis shows bias in modelled permafrost areas, reflecting bias in soil temperature. The significant correlation in Fig. 2a shows that, as expected, any bias in air temperature will propagate into soil temperature in the models. The slope being higher than 1 simply means that, among the CMIP5 models, those who have a cold bias in air temperature tend to simulate a larger $A_p/A_{MAAT<0}$. This is probably a pattern by chance, because $A_p/A_{MAAT<0}$ is largely determined by soil thermal

diffusivity in these models (Fig. 2b), a feature in the land surface component of the CMIP5 Earth system models, whereas bias in air temperature is a feature in the atmospheric component. The slope does not seem to have a meaningful implication, so we did not give the regression equation in Fig. 2a. Anyhow, we have now provided the data underlying Fig. 2 so that readers could easily derive the regression equation.

3. The form of the fitted linear equations needs to be consistent in figures, with significant levels given.

We have now added the p-values in figures that contain a correlation analysis.

“The form of the fitted linear equations needs to be consistent” - we suppose this refers to the regression equation in Fig. 2a, in which case, please see our response above. We checked that in other figures, the fitted equations are given in figure captions in a consistent form.

4. Why is thermal diffusivity given in form of log10 but not of real values in figures? The real values are more readable for readers.

We revise Fig.1 and Fig.2b to display the original D values at a log-axis, as shown below. Note that the regression equations are still derived after log-transform, as suggested by a previous reviewer in the first round, due to the log-normal distribution of the original D values.

Figure 1

Figure 2b

This is a very interesting paper and presents a unique approach to discuss the controls of soil organic matter on soil thermal dynamics in the northern high latitudes with modelling methods. This topic is valuable and seems to be important for earth surface research in cold regions. The revised version is much improved over the previous one. I would recommend that minor revision to be undertaken.

I don't think the authors answer Question 2 very reasonably. I'm afraid that I could not agree with the authors' answer about Fig. 2(a). The closer to the North, the higher the continuity of permafrost region in the North Hemisphere. Therefore, it causes the slope is bigger than 1.0. This seems to be the most important reason for the bigger than 1.0 slope, although I don't deny the technical problems for this.

Reviewer #1 (Remarks to the Author):

This is a very interesting paper and presents a unique approach to discuss the controls of soil organic matter on soil thermal dynamics in the northern high latitudes with modelling methods. This topic is valuable and seems to be important for earth surface research in cold regions. The revised version is much improved over the previous one. I would recommend that minor revision to be undertaken.

I don't think the authors answer Question 2 very reasonably. I'm afraid that I could not agree with the authors' answer about Fig. 2(a). The closer to the North, the higher the continuity of permafrost region in the North Hemisphere. Therefore, it causes the slope is bigger than 1.0. This seems to be the most important reason for the bigger than 1.0 slope, although I don't deny the technical problems for this.

We would like to thank the reviewer for the positive evaluation of our study.

Regarding Question 2, namely, why the slope in Fig. 2a is higher than 1, we agree that it could be related to the continuity of permafrost. When climate gets warmer, the area of continuous permafrost may decrease more strongly than the discontinuous permafrost. In CMIP5 models, without subgrid-scale soil temperature, the models are binary at each grid cell. Thus, models with a cold bias (a larger area of $MAAT < 0^{\circ}\text{C}$) produce more 'certainly' continuous permafrost, whereas warmer models may simulate no permafrost in southern grid cells where in reality there is discontinuous permafrost.

We therefore add the sentences at Line 165: "...with a standard deviation of 2.7 million km^2 (Fig. 2a). As expected, simulated near-surface permafrost area (A_p , defined as modelled active layer thickness less than 3 m (ref. 37), see Methods) is significantly correlated with $A_{MAAT < 0}$, with a slope slightly larger than one, which might be related to the continuity of permafrost (namely, in models with a warm bias, permafrost area is underestimated more strongly, as they miss the sub-grid discontinuous permafrost in the southern grid cells)."